# Metal Nanoparticles and Carbon-Based Nanomaterials for Improved Performances of Electrochemical (Bio)Sensors with Biomedical Applications

**DOI:** 10.3390/ma14216319

**Published:** 2021-10-22

**Authors:** Luminita Fritea, Florin Banica, Traian Octavian Costea, Liviu Moldovan, Luciana Dobjanschi, Mariana Muresan, Simona Cavalu

**Affiliations:** 1Faculty of Medicine and Pharmacy, University of Oradea, 10 P-ta 1 Decembrie, 410087 Oradea, Romania; fritea_luminita@yahoo.com (L.F.); florinbanica1@gmail.com (F.B.); marianamur2002@yahoo.com (M.M.); simona.cavalu@gmail.com (S.C.); 2Advanced Materials Research Infrastructure—SMARTMAT, University of Oradea, 1 Universitatii Street, 410087 Oradea, Romania; tcostea@uoradea.ro; 3Faculty of Electrical Engineering and Information Technology, University of Oradea, 1 Universitatii Street, 410087 Oradea, Romania

**Keywords:** metal nanoparticles, carbon-based nanomaterials, screen-printed electrodes, electrochemical (bio)sensors, biomedical applications

## Abstract

Monitoring human health for early detection of disease conditions or health disorders is of major clinical importance for maintaining a healthy life. Sensors are small devices employed for qualitative and quantitative determination of various analytes by monitoring their properties using a certain transduction method. A “real-time” biosensor includes a biological recognition receptor (such as an antibody, enzyme, nucleic acid or whole cell) and a transducer to convert the biological binding event to a detectable signal, which is read out indicating both the presence and concentration of the analyte molecule. A wide range of specific analytes with biomedical significance at ultralow concentration can be sensitively detected. In nano(bio)sensors, nanoparticles (NPs) are incorporated into the (bio)sensor design by attachment to the suitably modified platforms. For this purpose, metal nanoparticles have many advantageous properties making them useful in the transducer component of the (bio)sensors. Gold, silver and platinum NPs have been the most popular ones, each form of these metallic NPs exhibiting special surface and interface features, which significantly improve the biocompatibility and transduction of the (bio)sensor compared to the same process in the absence of these NPs. This comprehensive review is focused on the main types of NPs used for electrochemical (bio)sensors design, especially screen-printed electrodes, with their specific medical application due to their improved analytical performances and miniaturized form. Other advantages such as supporting real-time decision and rapid manipulation are pointed out. A special attention is paid to carbon-based nanomaterials (especially carbon nanotubes and graphene), used by themselves or decorated with metal nanoparticles, with excellent features such as high surface area, excellent conductivity, effective catalytic properties and biocompatibility, which confer to these hybrid nanocomposites a wide biomedical applicability.

## 1. Introduction

The development of sophisticated and miniaturized devices for sensing a broad range of biological molecules emerged as an imperative strategy for real-time diagnosis of different diseases and effective management of disease progression, based on rapid diagnostics, smart data analysis and statistical informatics analysis. The fabrication of smart devices based on bio-nanomaterials are considered the result of a transdisciplinary work from the materials science to the medical field, as the target analytes recognized by these devices are proteins, enzymes, antibodies, DNA/RNA probes, microorganisms, which can be detected with high sensitivity, accuracy and low detection limits [1]. The bio-detection system can be designed as bio-catalytic or bio-affinity-based system, according to the biochemical mechanism involved in the recognition: In the first case, the bioreceptor (proteins, enzymes, cells) undergoes a catalytic reaction with the analyte, while in the second case, a specific binding mechanism of the bioreceptor (aptamer, antibody) and analyte leads to an equilibrium [2]. According to the transduction pathways, the biosensors can be classified as: electrochemical, piezoelectric, optoelectronic and calorimetric.

The concept of nano(bio)sensors is related to nanostructures which are often incorporated into the (bio)sensor by attachment to a suitably modified platform. Nanoparticles and nanomaterials offer excellent properties for designing sensing systems with enhanced performances. Incorporating them in transducers, by attachment to a suitable modified platform, a high surface area can be achieved by tailoring their size and morphology, and hence, producing (bio)sensors with greater sensitivity and shorter response time. Hybrid materials and nanocomposite structures consisting of metallic NPs, combined with particular conductive polymers and a modified electrode, have been designed for electrochemical sensing, owing to their unique combination of biocompatibility, large surface area, and good conductivity [3,4]. The output of the sensing nano-platform can be connected to wireless devices for signal processing on a smart phone. Hence, these nano-devices could be easily handled and used as implantable devices in the body for health monitoring.

A nanomaterial is defined according to the European Commission as “a natural, incidental or manufactured material containing particles, in an unbound state or as an aggregate or as an agglomerate and for 50% or more of the particles in the number size distribution, one or more external dimensions is in the size range 1 nm–100 nm. In specific cases where warranted by concerns for the environment, health, safety or competitiveness, the number size distribution threshold of 50% may be replaced by a threshold between 1% to 50%” [5].

The original, innovative and complex configuration of this review combines the theoretical general aspects about metal nanoparticles, carbon-based nanomaterials and SPEs with their biomedical applications including the actual COVID-19 pandemic. The evolution from printable towards wearable sensors is also emphasized as the most recent and modern generation of sensing systems in healthcare. This review is focused on the electrochemical nano(bio)sensors based on screen-printed electrodes (SPEs), in which transduction is based on electrochemical techniques, while metal NPs and carbon-based nanomaterials are used as modifiers to improve the analytical performances. Each type of nano(bio)sensor is described in terms of the constituent nanostructures mentioning the large range of biomedical targets such as pathogens, cancer biomarkers and other relevant biomolecules, and also pharmaceutical compounds, analysis performed in the traditional laboratories or in-to-the-field (in-situ detection). Not the least, considering that intelligent diagnostics tools are urgently required to manage the COVID-19 pandemic, we present in this review recent approaches related to nanomaterials and SPEs in COVID-19 diagnosis, prevention and therapy. Some recent advances concerning the wearable sensors as non-invasive diagnostics tools for personalized healthcare management are mentioned.

## 2. Metal Nanoparticles

The capability to produce metal nanoparticles in the same size domain as proteins (1 to 100 nm) has led to a wide range of applications in the biomedical field. Their unique properties such as large surface to volume ratio and high percentage of atoms/molecules on the surface, can be exploited to improve the sensing and detection of several important biomolecules in healthcare-related fields. Metal NPs can be used alone and in combination with other nanostructures, with the aim of signal amplification, higher sensitivity and great improvements in the detection and quantification of different biomolecules. Actually, the role of metallic NPs in biosensing is directly related to their physico-chemical properties and changes that occur after binding the biomolecular analyte on the surface: (1) role as immobilizing platforms, (2) accelerating electron transfer, (3) catalyzing the chemiluminescent reaction with their substrates, (4) amplifying changes in mass, and (5) enhancing changes in refractive index [2].

Nanoparticles have physical and chemical properties that differ from those of the materials from which they are obtained. For example, metallic nanoparticles have a melting point significantly lower than that of the precursor metal: AuNPs melt at a temperature of around 300 °C (for 2.5 nm nanoparticles), compared to metallic Au whose melting point is 1064 °C [6]. Even the absorption of solar radiation is superior for NPs compared to thin metal sheets. AuNPs are also colored in solution from deep red to black. Moreover, metals are chemically inert in their macroscale form, while at nanoscale, their unique physicochemical features are remarkable. So far, a wide variety of metal/metal oxide nanoparticles has been synthesized for multiple applications: noble metals NPs (gold, silver and platinum NPs—used in biosensors, therapy, drug delivery, etc.), copper NPs, palladium NPs, lead NPs, selenium NPs, and metal oxide NPs (copper oxide, titanium dioxide, zinc oxide, indium oxide, iron oxide, etc.) (Figure 1) [7,8,9,10,11,12,13,14,15,16,17,18,19,20].

Different synthesis methods (bottom-up approach, top-down approach) have been developed in order to fabricate NPs with suitable features such as homogeneity, size and shape control, and versatile surface properties. Chemical, physical and biological fabrication route are currently employed, each of them with its own advantages and disadvantages (Figure 2).

Chemical methods include chemical or photochemical reduction, co-precipitation, thermal decomposition, hydrolysis and sol-gel method, these being the simplest methods used for preparation of metal nanoparticles. Usually, a metal precursor is reduced and in some cases a stabilizer is added to avoid NPs agglomeration. As a main advantage, desired size and shape nanoparticle can be prepared by this route, but on the other hand, the processes are difficult to control and there are drawbacks related to the reducing agents in terms of toxicity, poor reducing ability, high costs of reagents and impurities [13,21,22,23,24,25]. In the recent years, the electrochemical methods became a powerful tool for NPs synthesis due to their advantages: Facile, cost-effective, quick, highly efficient and environmental friendliness leading to NPs with high purity, controlled size, shape and composition (multimetallic NPs have been electrodeposited) by adjusting some parameters (potential, time, current density, number of scans, etc.) [26,27,28,29].

Physical methods include mechanical milling, grinding, vapor deposition, laser ablation, sputtering, spray pyrolysis, microwave irradiation, dissolution in supercritical fluids coupled with thermal reduction, etc. These are potentially clean techniques and the advantages consist in the possibility to produce thin metal films, while the nanoparticle properties such as surface morphology and crystal structure can be controlled. The disadvantages are generally related to the expensive instruments and specific operating requirements [30,31,32,33,34].

Biological methods are considered green synthesis methods, an emerging trend of nanotechnology, developed to overcome safety issues with respect to health and environment. For this purpose, plants, algae, fungi and different microorganisms (bacteria, viruses) are largely employed. The NPs size and morphology are influenced by the concentration of the biological partner, the growth phase of cells, the concentration of the metal ion, the pH of the solution, the temperature and the reaction time. These techniques have multiple advantages over other physical and chemical methods, such as cost effectiveness, eco-friendliness, and are easily scaled up for large scale production. Moreover, it does not involve sophisticated instruments or use of high pressure, energy, temperature and toxic reagents [19,25,35,36,37,38].

Concerning the biocompatibility of the metal NPs, the biological (and biotechnological) methods are considered to produce non-toxic, biocompatible and well-defined NPs. The use of various biomolecules for the NPs encapsulation was also an approach for increasing their biocompatibility and determining their mechanism of action, the modified and the spherical NPs presented lower toxicity [39,40]. The cellular uptake of metal NPs by endocytosis depends on their physico-chemical properties (type, size, shape, surface properties, dose). The biocompatibility assessment is achieved by using qualitative and quantitative analysis (based on cell staining and dye-conversion assays) [39].

The NPs characterization is achieved by various techniques such as: transmission electron microscopy (TEM), scanning electron microscopy (SEM), atomic force microscopy (AFM), X-ray diffraction (XRD), energy dispersive X-ray analysis (EDAX), X-ray fluorescence (XRF), Fourier-transform infrared spectroscopy (FTIR), ultraviolet-visible spectroscopy (UV–Vis), dynamic light scattering (DLS), Zeta potential analysis, electrochemistry, etc.

Various metallic nanoparticles are currently used in (bio)sensors fabrication, noble metals NPs such as gold, silver and platinum NPs being the most popular ones. Below we will focus on these three types of NPs, but also on other metal/metal oxide NPs also applied in (bio)sensing.

### 2.1. Gold Nanoparticles (AuNPs)

AuNPs, or colloidal gold, can be easily synthesized in sizes ranging between 3 and 100 nm in diameter, with different shapes. In a recent paper, De Souza et al. [41] presented a comprehensive review of the methodologies used in the synthesis of gold nanoparticles by chemical reduction, including the Turkevich method, synthesis with NaBH_4_ with/without citrate; the seeding-growth method; synthesis by ascorbic acid; green synthesis; the Brust–Schiffrin synthesis; and synthesis using other reducing agents. The most used method to synthesize quasi-spherical gold NPs is the chemical reduction of Au^3+^ and Au^+^ to Au^0^ ions using different reducing agent (sodium citrate, borohydrides, citric and oxalic acids, polyols, hydrogen peroxide, sulfites, etc.). HAuCl_4_ is the chosen precursor salt in most of the reported studies, in which gold is in the Au^3+^ oxidation state [42]. The use of a stabilization agent is necessary, often being the same molecule that of the reduction agent. The electrochemical synthesis of AuNPs is an often-employed method in the case of electrochemical sensor development where the optimization of the electrochemical method parameters influences the size (20–40, 30–80, 90 nm, deposition by cyclic voltammetry, amperometry) and the layer thickness of the electrodeposited NPs [10,43,44,45].

Nowadays, a special attention is paid to green synthesis methods, and therefore numerous research groups all over the world were focus on this field. The metal NPs biosynthesis from plants can be an efficient method for the development of a fast and ecological technology [37]. As a vegetal resource, amino acids, enzymes, flavonoids, aldehydes, ketones, amines, carboxylic acids, phenols, proteins and alkaloids can be successfully used to provide electrons to reduce Au^3+^ or Au^+^ into gold nanoparticles, the reaction being dependent on plant extract concentration, metal salt, reaction pH, temperature and incubation time. It is well known that the primary and secondary metabolites of plant are consistently involved in redox reactions of metabolic pathways. *Trigonella foenum-graecum, Hibiscus extract, Elettaria cardamomum, Garcinia cambogia, Areca catechu* and *Chenopodium album* are only few examples of plant extracts successfully employed to produce small AuNPs with diameter less than 20–30 nm [41,46,47,48,49,50].

Different types of bacteria were reported in the synthesis of AuNPs, either by intracellular or extracellular mechanisms, the production being initiated by tetrachloroaurate salt (AuCl_4_^−^). *Bacillus subtilis* 168 strain was successfully employed to intracellularly reduce Au^3+^ ions to AuNPs with the diameter range between 5–25 nm [30], while *Rhodopseudomonas capsulata* was found to successfully produce gold nanoparticles of different sizes and shapes: Spherical gold nanoparticles with diameter in the range of 10–20 nm were observed at pH = 7, gold nanoplates were observed at pH = 4 [51]. Other examples of microorganisms assisting AuNPs production, showing pH and temperature dependent mechanisms are: *Halomonas salina* (spherical NPs 30 to 100 nm in diameter) [52], *Delftia* sp. *strain KCM-006* (spherically shaped AuNPs 11.3 nm diameter) [53], *Bacillus subtilis 168* (octahedral 5–25 nm AuNPs inside the cell wall) [54], *Stenotrophomonas* sp. (multi-shaped 10–50 nm AuNPs, extracellular) [55], etc. In other studies, AuNPs and other metal NPs have been obtained by using fungi (*Aspergillus* sp., *Fusarium* sp. and *Penicillium* sp.) [56,57,58], viruses (*Tabacco mosaic virus*) and yeasts (*Pichia jadinii*).

It is generally accepted that the size, shape and function of the AuNPs are highly influenced by the physical and chemical parameters of their synthesis: the temperature of reaction, the stirring rate, the ratio of gold to reducing agent. In order to control the growth of the crystal nanostructures, surfactants are often used as surface coating in nanoparticles, as they have the ability to control the growth of nanocrystals to achieve desired morphologies], and hence, surfactant coated nanoparticles will remain well dispersed in relatively dilute solutions. For example, AuNPs protected by a compact shell of organo-thiols, are stable for long periods of time, with the possibility to be redispersed in organic solvents [59].

The biomedical applications of AuNPs are shape and size dependent. As emphasized in several research papers, gold nanoparticles are suitable to be used in electrochemical (bio)sensors design for a sensitive and selective detection of some important biomolecules, taking advantage of their good electrical conductivity and high surface area, which provides also a stable immobilization of various biomolecules retaining their bioactivity [7,10,60,61,62,63,64,65,66,67,68,69]. Other biomedical applications of gold nanorods and gold nanoparticles include to fight against cancer cells in photothermal therapy [8,70,71] and drug delivery [12,72].

### 2.2. Silver Nanoparticles (AgNPs)

The major routes of AgNPs preparation are physical, chemical, and biological synthesis, similar to AuNPs. Conventional physical methods for AgNPs fabrication are based on the evaporation–condensation approach and laser ablation technique, which are very efficient and moreover, they allow obtaining NPs with high purity, avoiding the use of potentially toxic reagents. The size, shape and yield of the AgNPs can be tailored by changing the parameters of the tube furnace (gas temperature, pressure) in the case of the evaporation-condensation technique, or laser power, duration of irradiation and liquid media selected in the case of laser ablation [73,74].

Chemical methods or wet chemistry usually employs three main components: metal precursors (metal salts), reducing agents (organic or inorganic), and stabilizing/capping agents. By this simple route, the nucleation and subsequent growth of AgNPs can be easily achieved. Silver nitrate AgNO_3_ is the most common salt used as precursor, while a large variety of reducing agents, such as sodium citrate, ascorbate, sodium borohydride (NaBH_4_), elemental hydrogen, Tollens reagent, N,N-dimethylformamide, poly(ethylene glycol)-block copolymers, thyo-glycerol, hydrazine, ammonium formate, etc., are used for the reduction of the silver ions (Ag^+^) in the aqueous or nonaqueous solutions [23,24]. Spherical AgNPs with diameter ranging from 10–100 nm were synthesized using ascorbic acid, sodium citrate, NaBH_4_, thiosulfate and polyethylene glycol as the reducing agents, while the surfactants such as citrate, polyvinylpyrolidone (PVP), cetyltrimethylammonium bromide (CTAB) and polyvinyl alcohol (PVA) were employed in order to stabilize particles and avoid sedimentation and agglomeration [75,76,77]. Optimized spherical and hemispherical AgNPs were obtained, having a diameter less than 10 nm, by adjusting four parameters: AgNO_3_ concentration, sodium citrate concentration, NaBH_4_ concentration and the pH of the reaction [78] Well-dispersed silver nanorods were reported by Ojha et al. [79], obtained by mixing the precursor solution of AgNO_3_ with citrate and adding NaOH or NaBH_4_ while stirring, the surfactant being cetyltrimethylammonium bromide. The aspect ratio (L/d) of the obtained nanorods (estimated from TEM) were 3.0  ±  0.1, 1.8  ±  0.1 and 1.1  ±  0.1, depending on the concentration of colloidal seed solution. Similarly, by adjusting the reaction conditions, including the ratio of PVP to silver nitrate, reaction temperature and seeding conditions, silver nanowires, with diameters ranging from 30 to 40 nm, and lengths up to ∼50μm, were reported when PVP was used as stabilizing agent [80]. Zhang et al. reported the fabrication of silver triangular bipyramids by photoinduced reduction of silver nitrate in the presence of sodium citrate, bis(p-sulfonatophenyl) phenylphosphine dihydrate dipotassium salt, adjusted by NaOH, under light irradiation (wavelength range between 500–650 nm) [81]. There are many examples in literature reporting the production of different shapes of silver nanoparticles synthesized with various chemical reductants, highlighting the advantages of this method, such as ease of production, high yield (contrary to physical methods, which have low yield), low cost. The main drawback remains however, the use of chemical reducing agents is harmful to living organisms. The electrochemical method (electrolysis) has been also employed for AgNPs synthesis using sacrificial anode (Ag electrode) and AgNO_3_ as electrolyte or precursor leading to AgNPs with size depending on the current density (10–50, 10–20, 20–80, 10–30 nm) [28,82,83,84].

Biologically-mediated synthesis of AgNPs emerged as a valuable option in order to overcome the shortcomings of chemical methods, and similar to AuNPs biosynthesis, bacteria, fungi, plant extracts, and small biomolecules were used as biological precursors in the context of environmentally friendly approaches. Examples of currently reported bacterial strains and fungi used for AgNPs synthesis are: *Bacillus amyloliquefaciens* [85], *Acinetobacter calcoaceticus* [86], *Pseudomonas aeruginosa* [87], *Escherichia coli* [88], *Brevibacterium casei* [89] and *Aspergillus sp., Fusarium sp., Penicillium sp.,* respectively [56,90,91]. On the other hand, plant extracts (*Aloe vera, Cocos nucifera, Ocimum tenuiflorum, Vitis vinifera, Chenopodium album*) rich in bio-compounds such as polysaccharides, tannins, saponins, phenolics, terpenoids, flavones, alkaloids, proteins, enzymes, vitamins and amino acids are largely available source of natural reagents for reduction processes involved in AgNPs production [50,92,93,94,95]. Overall, the biological methods demonstrated a controlled particle size, shape and mono-dispersity, while reducing time of preparation.

Similar to the AuNPs, AgNPs used in electrochemical (bio)sensors elaboration may improve the limit of detection of target molecules due to its high thermal, chemical stability, electrical conductivity, and catalytic activity [96,97,98,99,100,101,102]. Concerning other biomedical applications, AgNPs also presents a strong antimicrobial activity [103,104].

### 2.3. Platinum Nanoparticles (PtNPs)

PtNPs are also produced by biological or synthetic methods to be used in the biomedical field [105]. The physical methods include evaporation and condensation, laser ablation, solvothermal processes, all of them with advantages and disadvantages: high speed and no use of toxic chemicals, purity, uniform size and shape, versus low productivity, high cost, energy consuming, less thermal stability and high amount of waste. Chemical synthesis techniques include the sol–gel process, pyrolysis, microemulsion, hydrothermal, polyol synthesis and plasma chemical vapor deposition, in which metal precursors, capping or stabilizing agent, and reducing agent are the basic requirements. PtCl_2,_ H_2_PtCl_6,_ Pt(NH_3_)_4_(NO_3_)_2_ are the preferred precursors, while reducing agents like ascorbate, sodium borohydride, potassium bitartrate and trisodium are often used for the reduction process in order to tailor the NPs size and shape [106]. The biological synthesis of PtNPs was performed using different plants extracts (for example *Ocimum sanctum, Pinus resinosa, Fumariae herba*), while *Desulfovibrio desulfuricans* and *Acinetobacter calcoaceticus* were reported as bacterial routes to produce PtNPs by the reduction of Pt(IV) ion into Pt(0) NPs [107]. In this case, the biosynthesis route offer advantages such as small size (2–3.5 nm), monodispersion, no toxicity, cost effectiveness, rapid synthesis and environmental friendliness, but also drawbacks, being a laborious method, with relatively high cost, and less control over the NPs size and shape. The electrochemical technique (cyclic voltammetry, amperometry) was also employed for PtNPs synthesis [43,108,109]. Despite the high cost associated with its rarity in nature, PtNPs have various analytical applications (biosensors, fuel cells etc.) [110,111,112,113].

### 2.4. Other Metal/Metal Oxide Nanoparticles

Other metal NPs with attractive properties are: Palladium NPs and various binary and ternary combination of metal NPs as oxides, sulfides or metallic form (Bi_2_S_3_, ZnO, CuO, Co_3_N, Co_3_O_4_, Ni/ZnO, MoS_2_, IrO_2_, NiO, TiO_2_, PbS, Pd@Pt, Ni-Co@Pt, etc.) with different morphology (nanowires, nanosheets, nanofibers, nanoflakes, nanotubes, nanorods, core-shell) [114,115]. They can act as electrocatalyst (bimetallic and trimetallic NPs have synergic effect), but also are widely used as electrode modifiers with bioanalytical applications [116,117,118,119,120,121]. A special attention is paid to TiO_2_ nanomaterials due to some advantages: Ti is a biocompatible and abundant material, TiO_2_ nanomaterials are chemically stable, mechanically strong, highly uniform, having large surface area, with photo-catalytic properties and multi-functionalities, therefore TiO_2_ is often used as a supported material for decoration with other metal NPs with many biomedical applications [17,122,123,124].

Another important class is the magnetic nanoparticles, which include metal oxides, pure metals and magnetic nanocomposites, having various diagnostic and therapeutic applications in biomedical field [125]. Those containing iron oxides coated in polymer are the most used in bioanalysis due to their lower toxicity. Magnetic NPs are a powerful tool used in the design of electrochemical immunosensors due to their properties: They can be separated with an external magnet and then redispersed benefic for analytes separation and concentration, they have a large surface area leading to increased immobilization of biomolecules and facilitating the antigen-antibody reactions, controllable size, easy functionalization, superparamagnetic behavior, good conductivity and good biocompatibility [124,126]. A wide range of biomedical compounds have been analyzed by using immunosensors based on magnetic beads with increased sensitivity and reduced matrix effect [119,120,121,124,125].

The plenty advantages concerning the unique physical and chemical properties of metal NPs corroborated with their nanometer size are responsible for their wide range of biomedical applications (imaging, diagnostics, therapeutics). There still remains some challenges which should be addressed: Synthesis of the same size, agglomeration to be avoided, uniform distribution on the electrode surface and safe and efficient in vivo applications.

## 3. Carbon-Based Nanomaterials

The extensive family of nanoscale carbonaceous materials includes: Carbon nanotubes (CNTs), graphene, carbon-based quantum dots, fullerene, carbon black (CB), carbon nanowires, carbon nanofibers (CNF), carbon nanoribbons, carbon nanohorns, carbon nanocones, nanodiamonds, carbon nanoonios and mesoporous carbon [127,128,129,130,131]. These carbon-based nanomaterials are classified as 0D, 1D and 2D materials according to their shape with their representative members: fullerene (0D), CNTs (1D) and graphene (2D) (Figure 1). All the nanomaterials based on carbon possess some notorious inherent properties, such as high electrical conductivity, chemical stability, mechanical strength, high surface-to-volume ratio and biocompatibility [127,128,132,133]. They can be easily functionalized through covalent and non-covalent modification with functional groups or substances and also, they can be combined with other (nano)materials leading to hybrid (nano)composites with synergic effects for the envisaged applications. Their further functionalization with NPs not only drastically improves their physicochemical properties, but also prevents their agglomeration [133].

Their unique features concerning the electrical, optical, mechanical and thermal properties opened the way for a vast variety of applications, such as biosensing, bioimaging, cancer therapy, tissue engineering, drug delivery, biofuel cells, energy generation, storage, etc. [119,120,121,126,130,134,135,136,137,138,139,140,141,142,143].

Among all the nanostructured carbon materials, CNTs and graphene are the most widely studied, synthesized, functionalized and used for various analytical applications. The starting point for their synthesis is the graphite. As an amorphous and disordered product (concerning its structure), the graphite is stable at normal values of temperature and atmosphere. As it undergoes a controlled heating process, the disordered carbon atoms have a higher thermal energy that makes them arrange in a thermodynamically stable phase until reaching the sublimation point of 3915 K. Depending on the working temperature, graphene and CNTs are obtained [144].

Concerning the characterization of the carbon-based nanomaterials, a wide range of techniques have been employed so far such as: Raman spectroscopy, FTIR, UV–Vis, energy dispersive spectroscopy (EDS), X-ray photoelectron spectroscopy (XPS), XRD, AFM, TEM, SEM, laser scanning microscopy, electrochemistry, conductivity measurements, etc. [128,129,136,145,146]

### 3.1. Carbon Nanotubes

Carbon nanotubes have been discovered in 1991 by Iijima consisting in 1D cylindrical tubes of sp^2^ hybridized carbon atoms in a hexagonal lattice with delocalized π electrons [129]. They are unique due to the strong intermolecular bonds between the alternating hexagonal rings leading to an agglomerated structure [147,148]. They can have different lengths, thicknesses and number of layers. Their diameter is in the nanometer scale, meanwhile their length can reach up to several millimeters, even centimeters, possessing a high aspect ratio [127]. CNTs can be classified into single-walled carbon nanotubes (SWCNTs) and multi-walled carbon nanotubes (MWCNTs) according to the number of rolled graphene layers. SWCNTs have a single wall in the form of an empty cylinder on the inside, meanwhile MWCNTs have concentric tubes inside. These nanomaterials present good mechanical strength (100 times stronger than steel), excellent conductivity (CNTs conduct heat and electricity similar to copper), excellent electrocatalytic ability (they enhance the electron transfer for proteins/enzymes) and low density (half of the aluminum density) [127,132]. Concerning their thermal properties, the thermal conductivity is higher along the nanotubes than across them, an increased number of defects negatively influences the thermal conductivity, and MWCNTs have a higher value of thermal conductivity than SWCNTs due to their multi-layers [149]. CNTs have paved the way to the discovery of graphene.

The most used methods for CNTs synthesis are arc plasma, laser method and catalyzed chemical vapor deposition (Figure 2) [134,150]. By arc plasma method, an electric current is applied between two electrodes in an inert gas atmosphere and the deposition of CNTs on an electrode is achieved by consuming the other electrode [151]. This method has the disadvantage of obtaining a complex mixture requiring an additional purification in order to separate the CNTs from the residual components [152]. The use of laser method for CNTs synthesis has led to higher yields. The CNTs were obtained by laser vaporization of graphite electrodes with a mixture of catalysts (Co:Ni = 50:50, 1200 °C, argon atmosphere). For an efficient purification of CNTs, a vacuum heat treatment was performed at 1000 °C for removing fullerenes [153]. Catalyzed chemical vapor deposition is the most used method of synthesizing CNTs by catalytic deposition of hydrocarbons (acetylene) over a metal catalyst (Co and Fe) [132,154]. The optimization of this process consisted in the choice of hydrocarbons, the control of the reaction conditions, as well as the continuous elimination of CNTs as they are formed.

In order to surpass their limitations (especially their insolubility and tendency of agglomeration) and to increase the CNTs performance in analytical applications, the CNTs functionalization is performed including some groups: covalent, defect group, non-covalent and endohedral functionalization [127]. The non-covalent functionalization (hydrophobic and π-π interactions) presents the advantage of not disturbing the conjugated system of the CNTs. They can be functionalized with different functional groups: -OH, -COOH, -NH_2_, -F, etc. without significantly altering their properties. [155,156]. Moreover, more active binding sites can be created on the surface of nanotubes making them more easily dispersible in various solvents. Some types of functionalized CNTs are soluble in water and in other highly polar solvents [157,158]. Additionally, for biological applications different biomolecules (lipids, proteins, etc.) can be attached preserving their structural and functional integrity in order to increase the detection sensitivity [159,160]. The drop casting method widely used for CNTs deposition (especially for the electrode modification) presents a non-uniform distribution and fragility, limitations which can be overpassed by using polymers [127,161]. Recently, another approach has been employed by preparing free-standing buckypaper electrodes suitable for portable and wearable sensors and biofuel cells [135,136,137,138,162].

The various biomedical applications of CNTs requires an evaluation of their biocompatibility and toxicity. So far it was demonstrated that CNTs support the growth of neurons and osteoblastic cells; the functionalized CNTs improved the neuronal cell adhesion, mitochondrial membrane potential and concentration of acetylcholine; the CNTs coated with polymers were applied for bone tissue engineering [163]. CNTs are also promising drug delivery vehicles due to their covalent and noncovalent conjugation with drugs and biomolecules, entering the cell through cytoplasmic translocation. Nevertheless, CNTs present lung and embryonic toxicity [163].

### 3.2. Graphene

Graphene was discovered in 2004 by Novoselov et al. being awarded with a Nobel Prize in Physics in 2010 [164]. This 2D carbon-based nanomaterial with sp^2^ hybridization consists in a single-atom-thick layer of defect-free carbon atoms in a hexagonal network with delocalized π electrons. It possesses unique properties such as: excellent electrical conductivity (six times greater than copper and 60 times greater than CNTs), huge surface area (double than SWCNTs), excellent mechanical strength (200 times higher than steel), high thermal conductivity, optical transparence and elasticity [127,134,145,146]. Its electronic properties are influenced by the number of graphene layers and of edge defects. Until now, graphene is the thinnest and the strongest material. It is considered the building block for other carbon nanomaterials: by stacking, it leads to 3D graphite, by rolling, it forms 1D CNTs and by wrapping, it results in 0D fullerenes [165,166].

The first method of obtaining graphene is mentioned by Novoselov et al. in 2004 by mechanical graphite exfoliation [134,167,168]. An improved method was presented by Stankovich et al. using graphite oxide exfoliation in water under ultrasonication for 24 h at 100 °C [169]. Chemical methods have also been used to chemically extract graphene sheets from graphite by exfoliation [170,171]. A commonly used method in the laboratory is Hummers’ method of obtaining graphene oxide (GO) by adding potassium permanganate to a solution of graphite, sulfuric acid and sodium nitrate [172]. This method has been improved by removing sodium nitrate and adding sodium persulfate which ensures the complete exfoliation of graphite obtaining suspensions of individual graphite oxide sheets. Besides, it is more environmentally friendly by eliminating the formation of gases such as carbon dioxide and dinitrogen tetra oxide [173,174,175]. Other synthesis methods are the epitaxial growth [176] and chemical vapor deposition (Figure 2) [177,178]. Graphene can be obtained with an almost perfect structure and excellent properties by using these three methods [179].

Compared to graphene, GO can be cheaply produced by high-yielding chemical methods and is highly hydrophilic due to oxygen atoms that increase the distance between layers and can be exfoliated in water at moderate ultrasounds [179]. The presence of oxygen functional groups (hydroxyl, epoxide, carbonyl and carboxyl groups) improves the hydrophilicity, stability and the anchoring of different (bio)molecules. The chemical reduction of GO in order to obtain reduced graphene oxide (rGO) can be done by adding liquid reagents such as hydrazine, dimethylhydrazine, sodium borohydride, ascorbic acid (AA) and iodic acid to an aqueous dispersion of GO [10,180,181,182,183,184,185,186]. The electrochemical reduction of GO was also employed especially for electrochemical applications [127]. The difference between GO and rGO is the different percentage of oxygen content that leads to an insulating trend towards semiconductor behavior of GO, while rGO has a higher electrical conductivity [131,162,187]. Regarding the surface, rGO has a relatively larger specific surface area compared to GO [131,188]. Another modification of graphene structure is done by doping with heteroatoms (N_2_, B, S, P) being used for region activation with applications in electrochemical (bio)sensors [127,128]. Graphene and its derivatives (GO, rGO, graphene quantum dots) are important nanomaterials with outstanding properties which have been employed in various fields such as: biosensors, electronics, energy storage etc. [145,146,162,164,189,190]

Due to the high biomedical potential of graphene and its composites, the biocompatibility of these nanomaterials has to be evaluated aiming towards clinical translation. According to their physical and chemical properties, it was indicated that the nano-sized and surface coated materials were more biocompatible, meanwhile the micro-sized ones presented a high inflammation response, both in vitro and in vivo [191]. Additionally, lung toxicity was also reported. The impurities (toxic reagents used in synthesis) are responsible for cytotoxicity; therefore, the production of high-quality graphene is needed. The use of polymer nanocomposites reduces the toxicity favorizing the osteoblasts adhesion and proliferation [192].

### 3.3. Carbon-Based Quantum Dots

The latest discovered carbon-based nanomaterials are carbon dots (CDs—defined as carbon nanoparticles with less than 10 nm diameter) and graphene quantum dots (GQDs—possessing a graphene structure with layers less than 10 nm thick and 100 nm in lateral size), containing carbon core and many functional groups [193,194,195]. There are two main techniques concerning the synthesis of these quasi 0D nanomaterials: “Top-down” and “bottom-up” using characteristic precursors (carbonaceous materials; ethanol, nitriles, amino-acid) by physical, chemical and electrochemical methods [194,196,197,198]. Due to their unique structure, they present a lot of advantages, such as biocompatibility, nontoxicity, easy functionalization, chemical stability, abundant resources, low cost, versatility, attractive optical properties (photoluminescence), excellent electronic properties, high surface area and good solubility in many solvents [193,194,198]. They are widely applied in electrochemical sensing (as signal tags or as electrode modifiers alone or in combination with other nanomaterials), electrochemical flexible devices, electrocatalysis and biofuel cells [194,198,199,200]. GQDs have been more often used in electrochemical (bio)sensing due to their quantum confinement and edge effect, leading to their higher electrical and thermal conductivity besides their optical properties, as compared to the CDs and traditional quantum dots [193,197,201].

The wide range of carbon-based nanomaterials advantages generated by their unique properties combined with their nano-size are responsible for their widespread use in biomedical field. There still remains some disadvantageous issues that should be solved concerning synthesis and safety, such as expensive synthesis methods, removal of metal catalyst, impurities and other chemical reagents after the synthesis step, agglomeration, determination of proper toxicity for in vivo applications.

## 4. Metal Nanoparticles and Carbon-Based Nanomaterials in (Bio)Sensors Design

### 4.1. Screen-Printed Electrodes as (Bio)Sensing Platforms

According to the International Union of Pure and Applied Chemistry (IUPAC), a chemical sensor “is a device that transforms chemical information, ranging from the concentration of a specific sample component to total composition analysis, into an analytically useful signal” [202,203]. A biosensor, according to the IUPAC, can be defined as “a device that uses specific biochemical reactions mediated by isolated enzymes, immunosystems, tissues, organelles or whole cells to detect chemical compounds usually by electrical, thermal or optical signals” [202,204]. Meanwhile, an electrochemical biosensor is “a self-contained integrated device, which is capable of providing specific quantitative or semi-quantitative analytical information using a biological recognition element (biochemical receptor) which is retained in direct spatial contact with an electrochemical transduction element” [145,202]. Therefore, two basic units are primordial for the (bio)sensor construction: a receptor (recognition element—transforms the information into a signal) and a transducer (transfers the signal to a measured result). The electrochemical methods employed for the (bio)sensor analysis are various such as: amperometry, potentiometry, conductometry, voltammetry, impedance and surface charge sensing using field-effect transistors (FETs) [202]. The transducer part of an electrochemical sensor is an electrode. There is a wide range of electrodes (ion-selective, glass, gas, metal, carbon and chemically modified electrode), among which the carbon electrodes are maybe the most used [202]. The electrochemical methods have gained a great interest due to their advantages over others analytical techniques such as: sensitivity, selectivity, low cost, simplicity, ease of use, high reproducibility, low power requirement, real-time results and possibility of miniaturization and automation, features suitable for portable sensing devices with industrial and clinical applications.

In the recent decades, SPEs have attracted an increasing interest due to their advantageous characteristics, such as miniaturized form, great variety of electrode materials, use within a wide potential range, reduced sample volume, low cost, portable, faster time response and simplicity. The recent advances in technology have enabled the miniaturization of the electrodes and also of the potentiostats, the miniaturization and portability allowing to perform on-site and real-time analysis. SPEs are produced by screen-printed technology which consists in the deposition of more layers (ink, insulating material) on a substrate offering versatility in electrode design and electrode material, reproducibility, excellent uniformity, mass production, compatibility and modifications [205,206]. The process of SPE fabrication comprises of several steps: ink manufacture, stencil formation, layer printing and sinterization (thermal, photonic, plasma, microwave, electrical and chemical agents sintering) [5,207]. The SPE contains a three-electrode configuration: one or more working electrodes -WE, the pseudoreference electrode-PRE and the counter electrode-CE, with their connections on a chemically inert supporting material (various types: plastic polymer, paper, ceramic, alumina, etc.), having small dimensions (for example 3.5–5 × 1 × 0.05 cm for a classic SPE) (Figure 3) [5,205,206]. The dimension, the thickness and the form/shape can be controlled through the screen-printed technique. There are three main printing electrode technologies: Screen printing, inkjet printing and 3D printing, each one with their own advantages and disadvantages. Various nanostructured materials are used as embedded components (dispersions of conductive nanomaterials inks) in the WE’s configuration and/or as immobilized materials at the WE’s surface leading to nano-based printed or modified electrodes [207]. There are few disadvantages concerning the SPEs, such as laborious procedure for fabrication, variation in stability and reproducibility for lab-made and lab-modified SPEs, the need of an electrochemical pretreatment step to increase the reproducibility and a limited life-time for SPEs containing a bio-recognition element.

The composition and/or the surface of SPEs can be easily modified with a plethora of materials/substrates/substances in order to improve their analytical properties (especially the sensitivity and selectivity) and therefore their applications. The metallic nanoparticles and carbon nanomaterials are suitable for the SPEs modification because they are biocompatible, contribute to the biomolecules’ immobilization and increase the surface area, adsorption and conductivity of the electrode.

The metal nanoparticles employed in order to enhance the electrochemical signal, but also as labels in biosensors are: silver nanoparticles, gold nanoparticle, copper nanoparticles, iron nanoparticles, palladium nanoparticles, platinum nanoparticles and rhodium nanoparticles [206,208]. Their unique and excellent chemical and physical properties (high surface to volume ratio, high electron-transfer capability, high electrode conductivity) are responsible for the extensive use of NPs in (bio)sensors elaboration resulting in low limits of detection and anchoring platforms for biomolecules.

From the group of carbonaceous nanomaterials, the following can be mentioned: carbon black, carbon nanotubes, graphene, fullerene, carbon nanofibers, carbon nanohorns, carbon and graphene quantum dots with their unique electrocatalytic properties leading to sensitivity enhancement [5,206,208]. Carbon-based nanomaterials have brought uncountable benefits for electrochemical (bio)sensors due to their outstanding and attractive features such as: large specific surface area, elevated conductivity, high adsorption capability, decreased over-potentials, thermal conductivity, mechanical strength, high elasticity and functionalization possibility.

Composites based on metallic and metal oxide nanoparticles (monoatomic, polyatomic), carbon nanomaterials and polymers have been also envisaged as nanostructures in the SPE design [206,208]. The post-printing methods used for SPEs modification with nanomaterials includes: drop casting, electrodeposition, electrospraying, electrospinning, the Langmuir–Blodgett and the Langmuir–Schaefer methods [205,208,209,210].

The screen-printed electrochemical sensing platforms have been applied for sensing and monitoring of a wide range of target analytes in many fields, such as food and drinks, environmental analysis, pathogens, cancer biomarkers and other relevant biomolecules, pharmaceutical analysis and biological analysis, performed in the traditional laboratories and also in-to-the-field (in-situ detection) [43,119,120,121,205,206,208,209,210,211,212,213]. In the case of the biosensors’ elaboration, the SPEs have been modified also with biological elements such as enzymes, antibodies and nucleic acids by applying various methods of immobilization (casting, physical adsorption, electrochemical coating or inclusion into the ink) [5,206,210]. In the last years, a special attention was attributed to the detection of viral pathogens, such as the Human Immunodeficiency Virus, the Hepatitis viruses, the Zika Virus, the Dengue virus and SARS-CoV-2, using the combination between SPEs and nanomaterials for the development of portable, easy-to-use and cost-effective biosensors [214].

### 4.2. Nano(Bio)Sensors Based on Screen-Printed Electrodes with Biomedical Applications

#### 4.2.1. Nano(Bio)Sensors Based on Screen-Printed Electrodes with Medical Applications

Diagnostic and monitoring are essential in healthcare requiring accurate, sensitive, selective and fast results, features which can be accomplished by using screen-printed electrodes. These small size disposable electrodes can be easily integrated in point-of-care devices which are easy-to-handle, low-cost, portable and miniaturized sensors. Some examples of screen-printed electrodes modified with nanomaterials and applied for the determination of some biomolecules with great significance in the medical field are presented below and summarized in Table 1.

Exfoliated graphene oxide was electrochemically reduced by the potentiostatic method (the color changed from yellow-brown to black) on SPE doped with ionic liquid, presenting a uniform surface topography. This nanomaterial promoted the oxidation of NADH on this modified electrode by increasing the electron transfer: a negative shift of the oxidation peak potential of 0.22 V, a double increase of the peak current, a good resistance to fouling induced by a high density of edge-plane-like defective sites on carbon materials and a peak separation for ascorbic acid and NADH oxidation of 220 mV, in comparison with the non-modified electrode. The same superior electrocatalytic activity was also recorded for H_2_O_2_ analysis: oxidation started at +0.45 V and reduction at 0 V. Glucose oxidase was immobilized onto this modified SPE by cross-linking with glutaraldehyde leading to a nanobiosensor with improved analytical performance for glucose determination at −0.2 V. A good selectivity of the 3 sensors was demonstrated in the presence of ascorbic acid, uric acid and dopamine as interferents [215]. Layer by layer method was used for the SPE modification with MWCNTs, AuNPs and electropolymerized polyneutral red by optimizing the amount of CNTs and thickness of the film, then the nanoplatform was investigated toward the electrooxidation of NADH, showing the highest anodic peak current and good analytical parameters for amperometric determination [216].

Carbon nanomaterials, such as CB, SWCNTs-COOH, GO and rGO, were employed for the modification of homemade SPEs by drop casting, being characterized by XPS, Raman spectroscopy, SEM and electrochemistry (cyclic voltammetry, amperometry, electrochemical impedance spectroscopy -EIS). These nanostructured platforms were furthermore tested for NADH, AA and cysteine sensing, exploiting their greatest advantage such as a wide potential window and a high electrochemically accessible area. In the case of NADH oxidation, the lowest potentials of 400 and 440 mV were recorded for SPE modified with CB and SWCNTs-COOH, respectively; for AA oxidation the highest current peaks and the lowest potential peaks (90, 150 mV) were detected by using SWCNTs-COOH and CB, respectively. The same two nanomaterials have been responsible for the best electrochemical behavior of cysteine (oxidation at 580 mV). Thus, the analytical performances of the CB and CNTs are similar, but CB offers some particular advantages, such as it is cost-effective, suitable to obtain homogenous and stable dispersion and mass-producible [217]. NADH quantification was also performed by using another nanocomposite based on ruthenium dioxide-graphene nanoribbon drop casted on homemade SPE. RuO_2_ NPs presented an average diameter of 2 nm forming homogenous mats on the graphene matrix [218].

A hybrid nanocomposite based on platinum NPs doping into graphene sheets@cerium oxide presented good synergistic effects when tested towards the electrocatalytic reduction of H_2_O_2_. Various techniques, such as XRD, FTIR, SEM, EIS, cyclic voltammetry and amperometry, were applied for the characterization of the nanomaterials and of the modified SPE. The size controllable PtNPs were prepared with different electroless plating times, the best electrochemical response was obtained for PtNPs of 100 nm obtained over 200 s, indicating a larger surface area with more available active sites, but if the time and subsequently the size increases, the NPs aggregate leading to a compact surface with decreased current. The sensor also presented reliable reproducibility, long-range stability and selectivity (when adding glucose, AA, dopamine (DA), uric acid (UA)) [219]. A similar nanocomposite consisting in silver NPs, rGO@cerium oxide was synthesized and used for SPE modification being characterized by the same techniques mentioned before. Controllable in-situ synthesis of AgNPs with different sizes was achieved by a solvothermal process with a reaction time of 1–4 h, indicating that the reduction of Ag+ adsorbed onto the GO matrix is facilitated by the GO which acts as a gentle reductant. The best response for H_2_O_2_ reduction was registered for SPE modified with the hybrid nanomaterial with a 2 h reaction time (AgNPs having a size of 30 nm) [220].

Other important small biomolecules with high clinical importance, such as AA, UA, DA, levodopa (LD) and glucose, have been determined by using nanosensors based on SPEs. Graphene nanoribbons (obtained by chemical oxidation of CNTs) are a promising candidate for electrochemical sensors with the following advantages: Excellent electrocatalytic effect, enhanced faradaic currents and increased resistance to passivation leading to improved selectivity, sensitivity and reproducibility. SPE modified with reduced graphene nanoribbons (14% wt. oxygen content) presented the best electroanalytical performance for AA, LD and UA sensing in comparison with other related carbon nanomaterials (MWCNTs, oxidized graphene nanoribbons). This nanosensor allowed the simultaneous detection of AA, LD and UA at +0.08, +0.27 and +0.9 V, respectively, due to the reduced graphene nanoribbons containing more defects and edge sites and to the removal of oxygen functionalities [221]. In another study, the simultaneous detection of AA (−120 mV), DA (10 mV) and UA (220 mV) was achieved by employing a SPE modified with rGO and AuNPs (the simultaneous electrochemical reduction of both nanomaterials on the SPE surface is faster and more convenient), which was included in a smartphone-based integrated voltammetry system. The graphene oxide sheets containing a lot of oxygen-containing functional groups can be easier modified with NPs acting as a nanoscale building block to develop nanosensors [222]. A novel flow-injection amperometric nanosensor was developed for the accurate detection of UA based on SPE modified with a mixture of carbon black and graphene oxide by drop casting. The nanocomposite (1:1) was characterized by microscopic techniques (SEM, TEM) revealing a uniform distribution of carbon black spherical particles (30–50 nm) on the graphene oxide sheets. The electrochemical experiments indicated that the nanocomposite presented the highest current response for UA oxidation in comparison with other nanomaterials due to their synergic effects concerning the enhanced conductivity and increased surface area [223]. A SPE modified with rGO, polyneutral red and gold NPs (using commercial solutions of the nanomaterials) was elaborated for the amperometric determination of DA with good sensitivity, selectivity, reproducibility, stability and recovery rates [224]. The simultaneous detection of two neurotransmitters (dopamine and serotonin) was successfully achieved by employing SPE modified with MWCNTs and AuNPs [225]. A nanocomposite consisting in electrochemically generated polypyrrole nanoparticles and AuNPs was elaborated for SPE modification, leading to an increase of the active surface area, and this nanoplatform was used for sensitive and selective determination of serotonin [226]. In another study, the same nanocomposite elaborated through electrochemical techniques was used for immobilization of interleukin-6 aptamer and tested for the detection of the target cytokine [227].

The entrapment method for glucose oxidase immobilization onto the SPE surface has been addressed using various nanomaterials, such as SWCNTs, MWCNTs, rGO, silver NPs, platinum NPs and some polymers (poly(1-vinylimidazole), Nafion, poly(3,4-ethylenedioxythiophene, polyvinyl-alcohol and poly(3-aminobenzoic acid)) [228,249]. The nanosized platform based on platinum NPs, rGO and poly(3-aminobenzoic acid) being elaborated by one-step electrochemical deposition was also successfully applied for the determination of H_2_O_2_ and cholesterol [228]. Other enzymes such as glutamate dehydrogenase and lactate dehydrogenase have been immobilized onto the SPEs surface by entrapping in a mixture of chitosan and MWCNTs, drop-coating on the previously CNTs modified SPE and using a mixture of MWCNTS, glutaraldehyde and bovine serum albumin, those biosensors being applied for a sensitive detection of glutamate and lactate, respectively [249]. An enzyme free glucose sensor was elaborated by a three-step in-situ synthesis method of highly porous 3D hetero Cu(OH)_2_@CoNi-LDH core–shell nanotubes on the SPE surface. Firstly the CuNPs were electrodeposited on SPE, then Cu(OH)_2_ nanotubes were formed followed by the growth of CoNi-LDH nanostructures [229].

Several combinations of nanomaterials have been tested for the detection of aminothiols (cysteine, methionine, glutathione, homocysteine) including AuNPs and MWCNTs (with some polymers such as polyaniline and Nafion), deposited on a screen-printed gold electrode. AuNPs have been synthesized by a green procedure based on sonocatalysis leading to a diameter between 5 to 12 nm. The best result was obtained on the configuration consisting in AuNPs drop casted on SPE which indicates that the NPs enhance the electrochemical response of the conventional electrode [230].

The nanomaterials can be also included in the ink used for the SPE fabrication. In a study, homemade flexible screen-printed graphene electrodes have been developed and connected to a smartphone-based system. As a control, screen-printed carbon electrodes modified with graphene/graphene oxide by drop casting were also elaborated. The screen-printed graphene electrode presented on its surface a graphene-like layer structure, which led to a higher electron transfer rate in the case of norepinephrine electrochemical analysis [231]. Therefore, the sensitivity of the smartphone-based electrochemical system was improved by graphene used as a constituent material of the electrode. This novel set-up is suitable for portable and wearable point-of-care devices.

A comparison between the electrochemical performances of MWCNTs and graphene has been analyzed in the case of bilirubin oxidation. CNTs have been covalently bonded to the electrode surface, which was previously functionalized with NH_2_ groups. Graphene oxide were electrochemically reduced to the SPE surface. Better electroanalytical parameters have been recorded in the case of SPE modified with graphene, explained by a more enhanced electron transfer rate and a higher surface area calculated by electrochemical measurements [232]. A paper-based sensor was fabricated using graphene and carbon ink for the WE printing. AuNPs were then electrochemically deposited resulting in a diameter of 50–70 nm and a uniform distribution. The NPs increased the electrode surface area and also the anchoring sites for the covalent antibody immobilization. Multiple steps have been employed for the electrode modification leading to the elaboration of this label-free paper-based immunosensor with improved analytical performances (good sensitivity, selectivity, stability, reproducibility, repeatability, recoveries) toward C-reactive protein detection [233].

Some biomarkers important for diabetes mellitus diagnostic and monitoring were also analyzed by using SPEs modified with nanosized materials. For insulin detection, SPE was modified with Nafion-MWCNTs, followed by the electrochemical pulse potential deposition of NiO NPs. The pulse electrodeposition method generated NPs with a diameter below 30 nm, prevented the NPs agglomeration and controlled the film thickness, leading to a good stability, excellent surface coverage and enhanced electrocatalytic activity. NiO NPs act as an active catalyst for insulin oxidation [234]. Another SPE modified with a nanocomposite of CNTs and NiCoO_2_ (8:4.5 ratio) using Nafion as a binder was elaborated for insulin detection [235]. Glycated hemoglobin was tested by using label-free aptasensors based on six types of commercial carbon-modified SPEs (bare carbon SPE and carbon SPE modified with graphene, GO, MWCNTs, SWCNTs, CNF). The aptamer was non-covalently immobilized by π–π stacking with the nanomaterials (physical adsorption). The best results were obtained by employing SWCNTs platform, followed by MWCNTs and CNF [236]. The same 6 types of commercial carbon-based nanomaterial-modified SPEs (mentioned above) have been used as platforms for covalent immobilization of survival motor neuron protein antibody. In this case, the CNF-based immunosensor offered the best analytical performance for the protein target detection [237].

An electrochemical genosensor based on polyaniline and AuNPs electrodeposited on the SPE surface has been elaborated through multiple steps. The polymer decoration with NPs changed its redox to neutral pH suitable for biological interactions and also increased the surface area and conductivity improving the electrochemical biosensing performance. This biosensor based on a dual signal amplified strategy was applied for the detection of *E. coli* DNA and *E. coli* cells [238]. For the label-free detection of Hepatitis B surface antigen, an immunosensor was developed based on home-made SPE modified with a AuNPs-CNT nanocomposite (10–30 nm) and AgNPs. The nanocomposite was used in order to increase the surface area, conductivity and the biomolecule immobilization. AgNPs played the role of a redox probe for the direct detection, but also improved the analytical response due to its nanosize [239]. Electrochemically exfoliated graphene functionalized with AuNPs were deposited layer-by-layer on SPE and applied for pyoverdine detection (a virulence factor secreted by *Pseudomonas aeruginosa*) [240].

Tumor biomarkers are associated in patients with tumor or carcinoma; therefore, their early and accurate detection is essential. For this reason, the electrochemical immunosensors have been intensively studied combining the sensitivity of the electrochemical sensor and the specificity of the immunoreactions, paying special attention to the portable and miniaturized devices. A label free impedimetric immunosensor was elaborated based on electrodeposited AuNPs (40–100 nm) on the SPE surface, subsequently modified with self-assembled monolayers. Monoclonal antibody anti-carcinoma antigen 125 (CA125) was immobilized on the nanostructured platform which was successfully applied for the detection of CA125 (=MUC16) [241]. The detection of the same CA 125 biomarker was recorded on another immunosensor based on SPEs modified with AuNPs (“popcorn” nanostructures with 566 nm) and PtNPs (140 nm). The influence of the two nanomaterials on the electrochemical parameters have been compared, and the PtNPs presented the highest surface area leading to a lower limit of detection [242]. Carcinoembryonic antigen was determined on an immunosensor based on a SPE modified in one-step preparation with GO (electrochemically reduced) and AuNPs (140 nm—deposited by the electrochemical method). The nanocomposite increased the electron transfer kinetic and also the effective surface area, leading to a sensitive detection of the target [243]. A SPE modified with CNTs and AuNPs (the nanomaterials being incorporated in the SPE composition by the manufacturer) has been applied for an amperometric subnanomolar detection of the p53 protein (a biomarker for urinary tract carcinoma). This competitive immunosensor presented some advantages such as selectivity, precision, easy fabrication and low cost, being a valuable diagnostic tool for urologic malignancies [244]. A simultaneous determination of two metastasis-related biomarkers (interleukin-13 receptor-α2 and cadherin-17) has been realized on a dual SPE (SPE with two working electrodes) modified with graphene quantum dots (having peroxidase mimicking activity) and MWCNTs, a hybrid nanomaterial combination used for sensitivity enhancement and also as nanocarriers of antibodies for amplification purpose [245]. This dual amperometric sandwich-type immunosensor presented some relevant advantages over the traditional ELISA method such as: simplicity, rapid time response, decentralized analysis, affordability and portability. A graphene-SPE modified with electropolymerized polyaniline was tested for human chorionic gonadotropin. The impedimetric immunosensor was capable of a picogram determination [246].

#### 4.2.2. Nanomaterials and SPEs in COVID-19 Diagnosis, Prevention and Therapy

The recent global outbreak of COVID-19 disease is a continuous threat for public health; therefore, sensitive, fast, low-cost, easy to implement, real-time response, portable and wearable point of care systems are crucial for the control and monitorization of this disease.

In addition, the nanotechnology community is able to significantly contribute and fight against coronavirus disease 2019 (COVID-19), as nanomaterials are well known to possess antiviral properties with role in the prevention, diagnosis, and treatment of virus disease. The mechanism is related to inactivation of the outer layer of the virus upon interaction with hydrophobic nanomaterials surface, which may inhibit or completely destroy the virus [250]. The spike protein located on the outer surface is easily attached by nanoparticles in the form of drugs, coatings or nano medicines in the form of vaccines. Moreover, as nanoparticles are the same size as proteins, they can enter cells to enable expression of antigens or directly target immune cells for release of antigens. In this case, lipidic formulations and polymeric nanoparticles can be used as carriers, as their size, morphology and charge are tailored according the particular situation [251,252,253,254,255,256]. In a very recent review, Ghaemi et al. [257] highlighted the most used nanomaterials/nanoparticles with high potential in therapy, prevention and detection of targeted virus proteins: carbon nanotubes, graphene and graphene oxide, fullerene, quantum dots, CuNPs, ZnNPs, AgNPs, TiO_2_ and chitosan NPs. They concluded that the inclusion of nanomaterials in biosensors offer more detection capability, stability, simplicity of design, reliability and affordability. For example, TiO_2_ NPs can be used for surface decontamination in combatting SARS-CoV-2, in different formulations, such as aerosol, paint, water or air treatment systems, based on the photocatalysis mechanism and the production of hydroxyl radicals due to the water molecules oxidation, which promote the disinfection activity of TiO_2_ NPs [258]. In terms of prevention, different types of facemasks were designed based on graphene oxide nanoparticles as breathable barrier layers, taking into account the advantage of their hydrophobic surface to inactivate virus. Various types of nanoparticles and nanofibers have been introduced in mask production chains to improve safety performance and confer enhanced antiviral properties [259]. Graphene as a coating material for textiles is advantageous due to its mechanical properties, being also fire resistant, UV protective and conductive [260].

Lab-on-chip based strategies for smart diagnostic and personalized COVID 19 management (miniaturized SASR-CoV-2 biosensors) can be achieved via selecting a specific Anti-SARS-CoV-2 virus protein antibody for selective and sensitive detection within 30–40 min of operation time [261]. In terms of detection, optical biosensors, electrochemical biosensors, piezoelectric biosensors and thermal biosensors have been developed for respiratory virus detection. A review conducted by Samson et al. [262] summarizes the existing status of current diagnostic methods, along with their advantages and limitations, and the advantages of biosensor-based diagnostics over the conventional ones for the detection of SARS-CoV-2. A special attention was paid to novel biosensors used to detect RNA-viruses including CRISPR-Cas9-based paper strip, nucleic-acid based, aptamer-based, antigen-Au/Ag nanoparticles-based electrochemical biosensor, optical biosensor and surface plasmon resonance biosensor. As a nanotechnological approach, the gene-editing technique was modified by including a biological sensor using a CRISPR-Chip coupled with a graphene-based field effect transistor that can detect nucleic acids in a very small concentration (1.7 fM) without any amplification and in a very short time (15 min), while the detection of COVID-19 infection can be performed in less than 40 min [263].

The antigen lateral flow detection of SARS-CoV-2 as a point-of-care approach is realized by using a membrane strip with two lines: first line for the antibody-Au NPs, and second line for the captured antibodies. Biological samples such as blood or urine samples can be applied on the membrane, while, based on the capillary mechanism, proteins are drawn across the strip and an antigen/antibody AuNPs complex is formed and immobilized, a pair of red or blue lines becoming evident [257,264].

SPEs were customized in various configurations for SARS-CoV-2 detection (some components of the virus itself) [248,265,266,267,268,269], but also for the detection of some biomarkers useful for Covid 19 diagnosis (C-reactive protein, interleukins, tumor necrosis factor alpha, interferons, glutamate, breath pH, lymphocytes, platelets, neutrophils and D-dimer) [270]. Nanomaterials such as AuNPs, calixarene functionalized graphene, Au@Fe_3_O_4_ nanocomposite (400 nm) and carbon black served as catalysts, bioreceptor or labels, but, also, they improved the electrochemical performance of the SPE-based sensors applied for the virus detection [248,265,266,267,268,269]. In the case of biomarkers sensing, several nanomaterials have been employed for SPEs modification, such as rGO, AuNPs and Ag/Pt-graphene nanocomposite [270]. Beside AuNPs, Fe_2_O_3_ and Fe_3_O_4_ NPs docking interactions with the key amino acids in the spike protein receptor-binding domain of SARS-COV-2 were demonstrated recently, revealing that the interactions are associated with conformational changes in viral structural proteins and subsequent inactivation of the virus [271]. In a recent paper, Mahari et al. [272] described the fabrication of an in-house built biosensor based on fluorine-doped tin oxide electrode and AuNPs coupled to a nCOVID-19 antibody, which was demonstrated to be very specific in the detection of the nCOVID-19 spike antigen. In terms of sensitivity, this immunosensor could detect the nCOVID-19 antigen in concentrations from 1 fM to 1 µM within 10–30 s.

In terms of therapy, in the context of urgent treatment development and limited benefit of Dexamethasone and Remdesivir, SARS-CoV-2-specific therapies to treat coronavirus disease has emerged. In a very recent paper, Adi Idris et al. [273] reported the development and screening of two novel liposomal formulations for the delivery of small interfering RNA therapeutics to the lungs. Small interfering RNA molecules are short double-stranded RNA molecules that encodes the genome of coronaviruses, and by encapsulation of these active molecules in cationic liposomes with average size of 80 nm, an encapsulation efficiency of 97.6% was achieved, followed by in vivo injection in mice. The assessment was performed after 24 h, showing the localization of fluorescence particles in the lung (21%), liver (67%), and spleen (12%). The authors demonstrated robust repression of virus in the lungs and a pronounced survival advantage to the treated mice compared to the control. Moreover, the authors highlighted that the treatment based on the nano-liposomal approach is scalable and can be administered upon the first sign of SARS-CoV-2 infection in humans, suggesting also that this therapeutic approach is useful as an adjunctive therapy to current vaccine strategies.

The general representation of screen-printed electrodes modified with nanostructured materials for biomedical applications is presented below in Figure 4.

#### 4.2.3. Nano(Bio)Sensors Based on Screen-Printed Electrodes with Pharmaceutical Applications

The monitorization of drugs concentration in body fluids, pharmaceutical formulations and other real samples is essential in order to optimize the therapy, to monitor the treatment and to reduce the risk effects, requiring simplicity, sensitivity, selectivity, real-time response, cost effectiveness, portability and miniaturization, advantages characteristic for electrochemical nano(bio)sensors based on SPEs. Pharmaceutical substances having a wide range of pharmacological activities have been detected on screen-printed electrodes modified with various nanosized materials by exploiting various electrochemical techniques (Table 2).

Vitamins’ analysis has been performed on different SPEs modified with carbon-based nanomaterials leading to an µM detection limit [249]. Super critical CO_2_ medium was used as a novel approach for the decoration of MWCNTs with metal nanoparticles, the best results for vitamin B6 determination was achieved by using the SPE modified with RuNPs-MWCNTs, resulting in a minimum three-fold increase in sensitivity [274]. A nanocomposite based on MWCNTs-polyvinyl chloride and calixarene as a molecular recognition element have been used for SPE modification and applied for gentamicin potentiometric determination improving the stability, response time, lifetime, sensitivity and selectivity of the sensor [275]. Carboxylated MWCNTs decorated with AuNPs using ethylenediamine as a cross-linker were drop-casted on SPE and applied for electrochemical determination of amoxicillin combining the good electron transfer property and catalytic property of the two nanomaterials [276]. AuNPs have been electrochemically deposited on SPE (with spherical shape and size between 13–58 nm) and applied for moxifloxacin hydrochloride detection with good performance [277]. A low-cost SPE modified with fullerene, reduced graphene oxide and nafion has been elaborated in order to evaluate metronidazole with high stability, repeatability, reproducibility, fast response and low cost, performance which is due to the electrocatalytic synergic effect of the nanomaterials [278]. Among the electrodes used for isoniazid and rifampicin electroanalysis, the SPEs modified with nanomaterials recorded good analytical results [279]. Fe_3_O_4_@polypyrrole-Pt core-shell nanoparticles have been synthesized and then drop casted on SPE allowing a simultaneous detection of two anticancer drugs (6-mercaptopurine and 6-thioguanine) [280]. A SPE was elaborated during three steps: anodic pre-treatment, drop cast of CeO_2_ NPs annealed at 900 °C and heat treatment in vacuum, then it was applied for diclofenac determination. The SEM image of the nanosensor indicated o smoother surface of the electrode because the NPs filled the graphite gaps. The NPs also increased the electrochemical active area of the working electrode [281].

Three different types of SPEs modified with CNTs, carbon nanofibers and graphene were tested for the simultaneous determination of paracetamol, ibuprofen and caffeine in water samples. The best results were obtained by using the carbon nanofiber electrode [282]. The investigation of SPEs (commercial and modified in the lab) based on carbon nanomaterials (CNTs and graphene) was achieved for another simultaneous determination of melatonin and serotonin. Concerning the electroactive area of both types of SPEs, the electrode containing SWCNTs presented a higher surface in comparison with the MWCNTs, meanwhile the graphene based SPE exhibited the highest one. The SPE modified in the lab with graphene reduced nanoribbons presented the highest electroactive surface area from all the analyzed electrodes. For the simultaneous determination of the two substances, it was applied the SPE modified with graphene oxide nanoribbons [283]. Another simultaneous detection of three antiallergic drugs (phenylephrine, chlorpheniramine, dextromethorphan) was performed by using a SPE drop-casted modified with manganese hexacyanoferrate/chitosan nanocomposite. The TEM images indicated manganese hexacyanoferrate nanocubes with a size of 162.9 nm homogenously distributed over the chitosan nanoparticles [284]. MWCNTs and crown ethers have been used for SPE modification leading to a potentiometric sensor applied for a sensitive, fast and simple determination of pioglitazone [285]. A SPE based on graphene quantum dots was elaborated by drop casting and applied for isoproterenol sensitive detection [286]. Some different SPEs modified with Gd_2_O_3_ NPs (size 23 nm, obtained via thermal precursor decomposition), La^3+^/Co_3_O_4_ nanocubes and Fe_3_O_4_@cellulose nanocrystal/Cu nanocomposite and graphene have been developed for the determination of two antidepressant drugs (venlafaxine and sertraline) [121,287,288]. Various nanomaterials used for SPEs modification have been also employed for cysteine and acetaminophen analysis with considerable performances [119,213].

## 5. Wearable Sensors in Healthcare—Some Recent Advances, Challenges and Perspectives

Although significant achievements have been realized in the development of nano-bio-sensors for point-of-care diagnostics including cancer, diabetes, malaria, HIV, it would be greatly desirable to explore a wider category of nanomaterials with superior properties to improve sensor performance for larger applications. The integration of nanomaterials in point of care testing and the opportunity of realizing portable, easy to use, cost effective, and miniaturized analytical devices represent a continuous challenge, including for personalized medicine. The next generation of point-of-care devices have to overcome the actual limitations related to inadequate detection sensitivity to distinguish biomarkers at the different stages of the diseases, while improving the selectivity at molecular level, for monitoring patient health at anywhere and anytime [289,290].

Taking into account that rigid materials could be converted to flexible ones, when their structures are reduced to nanoscale, wearable devices have revolutionized the healthcare system by decreasing the hospitalization time and by providing more reliable and timely information. Among other advantages are the comfortability and daily care possibilities. Flexible wearables, textile-based wearables, epidermal-based wearables, biofluidic-based wearables, or wearable drug delivery systems are only few examples. Wearables biosensing devices can be employed for different body parts, monitoring different psychological and physiological parameters (in saliva, blood, urine, sweat) that are crucial for the diagnoses or treatment purpose [291]. For example, in diabetes, which is among a group of underlying conditions with increased risk of severe COVID-19 disease, continuous glucose monitor devices are now commercially available as Dexcom G6 [292], with triple action: sensor, transmitter, and receiver. The automatic applicator- the sensor wire—is inserted under the skin, the readings are transmitted to the receiver and visualized in real time. Another similar approach, aiming to help diabetes monitorization and prevention of the complications, is the development of tears-based wearable device, commercially available as Triggerfish [293] that monitors the intraocular pressure of glaucoma patients for the diagnosis of diabetes. Challenges in monitoring cardiac patients may have substantial solutions by developing tattoo-based wearable device for ECG monitoring, which consists of miniaturized electronic components built on a graphene/PMMA bilayer substrate, being effective for monitoring different biopotentials like ECG, EMG and EEG signals [294,295,296].

As the wearables are tailored to an individual’s physiological responses, such as heart rate, electrodermal activity-responsible for the emotional status and skin temperature, these signals can be extracted using the autonomic nervous system and provide feedback to patients with neurological diseases, like epilepsy, Parkinson’s and Alzheimer’s disease [297]. Another future approach in sensing under routine and even sedentary activity can be used to develop sweat-based biomarker monitoring practical for daily life, in a convenient glove-based form, for rapid accumulation of natural thermoregulatory sweat without active sweat stimulation. The fingertips, palm, and back of hand possess some of the highest sweat gland densities on the body, being accessible sites for monitoring natural sweat, and hence, glove-based sensing platforms (nitrile gloves and finger cots) are attractive for in situ detection of diverse biomarkers, including electrolytes and xenobiotics [298]. In the context of new challenges related to the COVID-19 pandemic, a major new driving force should be directed toward the development of modern wearable medical devices, suitable to monitor temperature, heart rate, sleep quality, blood oxygenation, which are crucial parameters for the early detection of COVID-19.

## 6. Conclusions and Future Outlook

In this review, we aimed to highlight the great importance and recent development of effective diagnostic tools for early detection of clinical biomarkers, not only in terms of detecting disease, but also related to physiological signatures that are predictive of potential disease. This comprehensive review is focused on the main types of metal NPs and carbon-based nanomaterials used for electrochemical (bio)sensors design, especially screen-printed electrodes, with their specific biomedical applications, improved analytical performances and miniaturized form. A brief overview about metal NPs and carbon-based nanomaterials concerning their synthesis, unique and specific properties and their use as electrode modifiers have been summarized pointing out the medical and pharmaceutical applications of the nano(bio)sensors based on SPEs. Some recent advances in the area of two important and actual topics have been also emphasized: nanomaterials and SPEs involvement in the COVID-19 management and wearable sensors in healthcare.

Nowadays, there is a tremendous need for rapid analysis, continuous monitoring systems with high accuracy for biomolecular detection. Real-time diagnostic decision and rapid manipulation is crucial, mainly in the context of COVID-19 pandemic management. Nanotechnological approaches will extend the limits of currently employed (bio)sensors and, moreover, they will open a new window toward personalized medicine, offering new solutions to the main challenges in the diagnostic and therapeutic fields. Future research should focus on some improvements concerning the nanomaterials characteristics and the sensor design in order to enhance their performances with multi-disciplinary efforts. The real sample analysis with more enhanced sensitivity and selectivity is still a challenge for researchers aiming the validation of the electrochemical nano(bio)sensors in comparison with the traditional analytical procedures. The reproducibility is another key aspect which needs to be solved for large-scale production of electrochemical sensors and their introduction on commercial market. The miniaturized, portable or wearable sensors which can perform on-site and real-time analysis will gain tremendous importance at the commercial level, with a huge impact on the health system.

## Figures and Tables

**Figure 1 materials-14-06319-f001:**
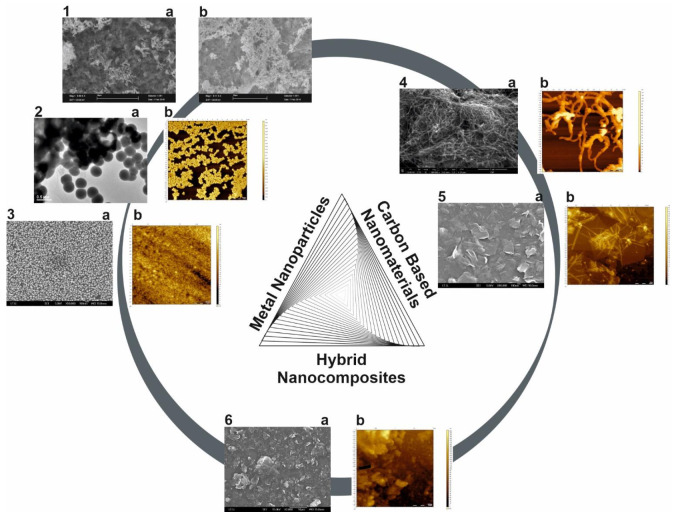
Microscopic images of some metal nanoparticles (1a—SEM image of TiO_2_ NPs, 1b—SEM image of TiO_2_ doped with SeNPs; 2a—TEM image of SeNPS; 2b—AFM image of SeNPs; 3a—SEM image of AuNPs, 3b—AFM image of AuNPs), carbon-based nanomaterials (4a—SEM image of MWCNTs, 4b—AFM image of MWCNTs; 5a—SEM image of graphene, 5b—AFM image of graphene) and hybrid nanocomposite (6a—SEM image of graphene + AuNPs, 6b—AFM image of graphene + AuNPs) (original images).

**Figure 2 materials-14-06319-f002:**
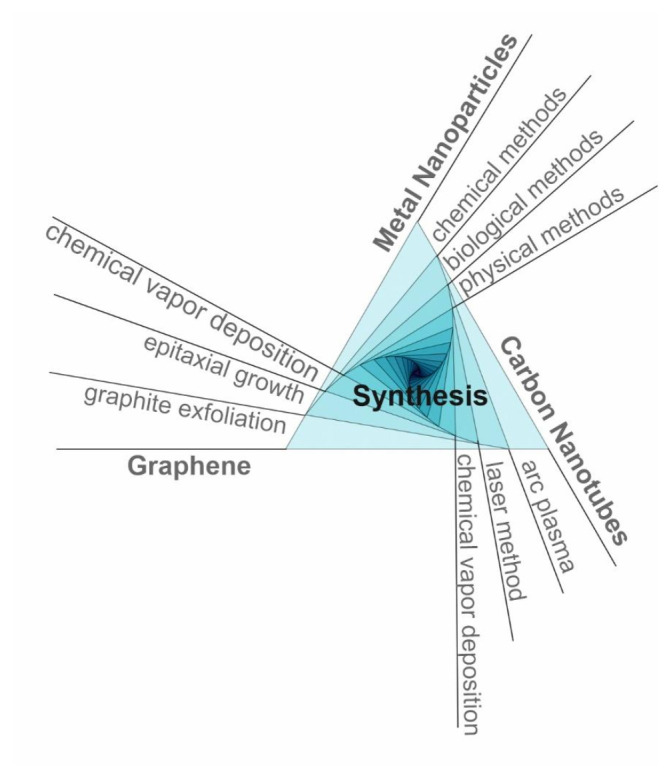
Schematic representation of main synthesis methods of metal NPs and carbon-based nanomaterials.

**Figure 3 materials-14-06319-f003:**
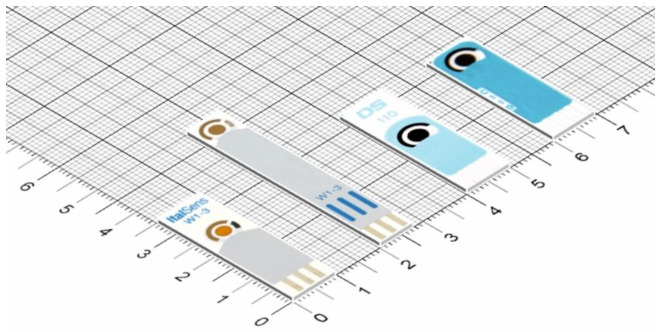
Some examples of commercially available screen-printed electrodes (from different manufacturers: https://www.dropsens.com/en/screen_printed_electrodes_pag.html; https://www.palmsens.com/products/sensors/screen-printed-electrodes/, accessed 10 September 2021).

**Figure 4 materials-14-06319-f004:**
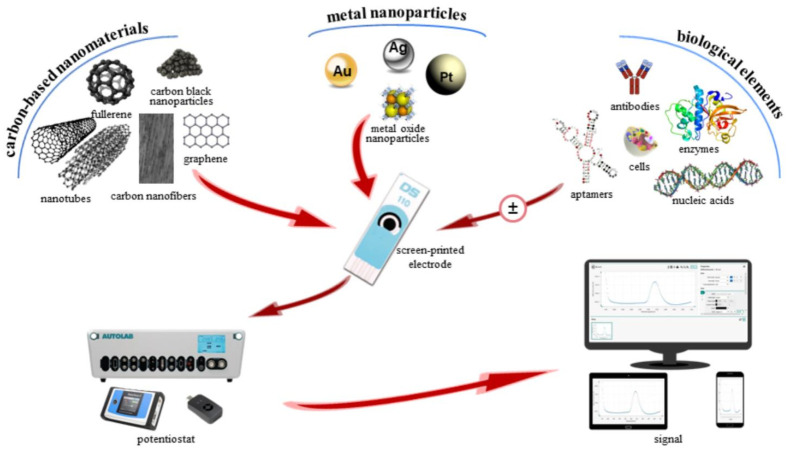
Schematic representation of nano(bio)sensors based on screen-printed electrodes: modification with carbon-based nanomaterials, metal nanoparticles, with/without biological elements and the electrochemical analysis.

**Table 1 materials-14-06319-t001:** Screen-printed electrodes modified with various nanomaterials for medical applications (some examples).

Type of NPs	Analyte	Method/ LOD/LR	Real Samples/ Recovery	Ref.
Reduced exfoliated graphene oxide	NADH	Amperometry		[215]
		2 µM/10 µM–2.0 mM		
	H_2_O_2_	Amperometry		
		0.05 µM; 0.08 µM		
		0.1 µM–2.0 mM;		
		0.15 µM–1.8 mM		
	Glucose	Amperometry	Milk samples	
		1 µM/5 µM–10.0 mM		
MWCNTs, AuNPs	NADH	Amperometry		[216]
		3.72 µM/12.4–150 µM		
Carbon black	NADH,	Amperometry		[217]
	Ascorbic acid, cysteine	1 µM		
Ruthenium dioxide-graphene nanoribbon	NADH	Amperometry 0.52 µM/1–1300 µM		[218]
Platinum NPs, graphene sheets@cerium oxide	H_2_O_2_	Amperometry	Contact lens clear solution/	[219]
		0.43 µM/1 µM–10.0 mM	99.5–102%	
Silver NPs, rGO@CeO_2_	H_2_O_2_	Amperometry	Contact lens clear solution/	[220]
		0.21 µM/0.5 µM–12 mM	100–103.5%	
Reduced graphene nanoribbons		DPV	Urine	[221]
	Uric acid	5 µM	97–101%	
	Levodopa	10–50 µM		
	Ascorbic acid	1–5 mM		
Reduced graphene oxide, gold NPs		DPV	Urine	[222]
	Ascorbic acid	1.04 µM/20–375 µM	95–98.89%	
	Dopamine	0.29 µM/1–160 µM		
	Uric acid	5.4 µM/25–200 µM		
Carbon black, graphene oxide	Uric acid	Flow-injection amperometry	Urine	[223]
		0.01 µM/0.05–2000 µM	95%	
Reduced graphene oxide, gold NPs	Dopamine	Amperometry	Blood	[224]
		0.17 µM/0.57–500 µM	101.5–102.5%	
MWCNTs, AuNPs		DPV	Serum, Tears, Saliva	[225]
	Dopamine	0.3 µM/1–100 µM	111.18%, 97.78%,	
	Serotonin	0.8 µM/2.5–100 µM	108.53%	
Polypryrrole NPs, AuNPs	Serotonin	SWV	Serum	[226]
		33.22 nM/0.1–15 µM	100.27–103.06%	
Polypryrrole NPs, AuNPs	Interleukin-6	EIS	Serum	[227]
		0.33 pg/mL/	101.41–102.45%	
		1 pg/mL–15 µg/mL		
Platinum NPs, reduced graphene oxide		Amperometry	Serum	[228]
	Glucose	44.3 µM/0.25–6.0 mM	82.2–104.1%	
	H_2_O_2_	5.24 µM/0.01–0.80 mM		
	Cholesterol	40.5 µM/0.25–4.0 mM		
Cu(OH)_2_@CoNi-LDH core–shell nanotubes	Glucose	Amperometry	Blood	[229]
		6.7 µM/20 µM–8 mM	97.5%	
AuNPS		Amperometry (HPLC-EC)	Dietary supplements	[230]
	Cysteine,	3.1 µM/10–80 µM	97.25–99%	
	Methionine,	1 µM/3.3–30 µM		
	Glutathione,	0.1 µM/0.3–10 µM		
	Homocysteine	0.6 µM/2.2–30 µM		
Graphene (in the ink of SPE)	Norepinephrine	SWV		[231]
		0.265 µM/1–30 µM		
	Bilirubin	Amperometry	Blood	[232]
MWCNTs		0.3 µM/0.5–500 µM	94–106.5%	
rGO		0.1 µM/0.1–600 µM		
Graphene (in the ink of SPE), AuNPs	C-reactive protein	EIS	Blood	[233]
		15 ng/mL/0.05–100 µg/mL	97.9–103.9%	
NiO NPs, Nafion-MWCNTs	Insulin	Amperometry		[234]
		6.1 nM/20–260 nM		
CNTs-NiCoO_2_ in Nafion	Insulin	Amperometry		[235]
		1.06 µg/mL/0.17–75 µg/mL		
SWCNTs (commercial SWCNTs/SPE)	Glycated hemoglobin	SWV		[236]
		0.03 pg/mL/		
		0.1–1000 pg/mL		
Carbon nanofiber	Survival Motor Neuron Protein	SWV	Whole blood	[237]
		0.75 pg/mL		
		1 pg/mL–100 ng/mL		
PANI/AuNPs	*E. coli* DNA	CV	Urine	[238]
		0.5 fM/1000–0.001 pM		
	*E. coli* cells	4 CFU/mL		
		4 × 10^6^ CFU/mL		
AuNPs-CNTs, AgNPs	Hepatitis B surface antigen	DPV	Blood	[239]
		0.86 ng/mL/1–40 ng/mL	80.70–91.40%	
Graphene, AuNPs	Pyoverdine (Pseudomonas aeruginosa)	DPV	Serum, saliva, tap water	[240]
		0.33 µM/1–100 µM	98.41–102.12%	
AuNPs	Carcinoma antigen 125	EIS	Blood	[241]
		6.7 U/mL/0–100 U/mL		
	Carcinoma antigen 125	EIS		[242]
AuNPs		419 ng/mL		
PtNPs		386 ng/mL		
		450 ng/mL–2.916 µg/mL		
rGO-AuNPs	Carcinoembryonic antigen	Amperometry, CV		[243]
		0.28 ng/mL/		
		0.5–50 ng/mL		
		181.5 ng/mL/		
		250–2000 ng/mL		
CNTs-AuNPs (in the ink)	p53 protein	Amperometry	Urine	[244]
		14 pM/20 pM–10 nM	91–132%	
Graphene quantum dots—MWCNTs		Amperometry	Cancer cells	[245]
	Interleukin-13	1.4 ng/mL		
	receptor-α2	4.92–100 ng/mL		
		0.03 ng/mL		
	Cadherin-17	0.11–10 ng/mL		
Graphene (in the ink), polyaniline	Human chorionic gonadotropin	EIS	Urine	[246]
		0.286 pg/mL/		
		0.001–50 ng/mL		
Calixarene functionalized graphene, Au@Fe_3_O_4_	SARS-CoV-2	DPV	Various biological fluids	[247]
		200 copies/mL	20–100%	
		3 aM/		
		10^−17^–10^−12^ M		
Carbon black	SARS-CoV-2	DPV	Saliva	[248]
		19 ng/mL (S protein)		
		8 ng/mL (N protein)		

**Table 2 materials-14-06319-t002:** Screen-printed electrodes modified with various nanomaterials for pharmaceutical applications (some examples).

Type of NPs	Analyte	Method/ LOD/LR	Real Samples/ Recovery	Ref.
Chemically reduced graphene oxide (in the ink)	Vitamin C	DPV/	Injection formula	[249]
		0.95 µM/4–4500 µM		
MWCNTs	Vitamin C	Amperometry/	Tablet, capsule	[249]
		11 µM/50–400 µM		
MWCNTs	Vitamin B6	Amperometry/	Tablet, capsule, drinks, cereal	[249]
		8 µM/25–300 µM		
		DPV/		
		1.5 µM/2–72 µM		
RuNPS-MWCNTs	Vitamin B6	LSW	Tablet, ampoule, drinks	[274]
		0.8 µM/2.6–200 µM	92–107%	
AuNPs	Vitamin B7	SWV/		[249]
		8.3 nM/0.01 nM–0.01 M		
		Amperometry/		
		14 nM/1 nM–1µM		
MWCNTs	Vitamin B9	Amperometry/	Tablet, capsule	[249]
		8 µM/50–400 µM		
MWCNTs	Gentamicin sulphate	Potentiometric titration	Ampoule, ointment, cream, surface water	[275]
		75 nM/0.1 µM–10 mM	97.5–101.3%	
Carboxylated MWCNTs-AuNPs	Amoxicillin	AdSV	Bovine milk	[276]
		0.015 µM/0.2–30 µM	91.5–95.5%	
AuNPs	Moxifloxacin hydrochloride	DPV	Urine	[277]
		11.6 µM/8–480 µM	99.8–101.6%	
Fullerene-reduced graphene oxide	Metronidazole	SWV	Serum, Urine	[278]
		0.21 µM/0.25–34 µM	92–100%	
BiO nanorods	Isoniazid	DPV	Serum	[279]
		1.85 µM / 5–100 µM	92–104%	
Fe_3_O_4_@polypyrrole-Pt core-shell nanoparticles		DPV	Urine	[280]
	6-mercaptopurine	10 nM/0.04–330 µM	Anticancer tablets	
	6-thioguanine	6 µM/0.1–400 µM		
CeO_2_ NPs	Diclofenac	SWV	Water samples	[281]
		0.4 µM/0.1–25.6 µM		
Carbon nanofibers		DPV	Tap water	[282]
	Paracetamol	0.03 mg/L	97.6–103.1%	
		0.09–0.8 mg/L		
	Ibuprofen	0.6 mg/L		
		2.2–10.2 mg/L		
	Caffeine	0.05 mg/L		
		0.2–1.1 mg/L		
Graphene nanoribbons	Melatonin	DPV	Tablet, capsule	[283]
		1.1 µM/0.005–3 µM	97.8–98.3%	
Manganese hexacyanoferrate/chitosan nanocomposite		SWV	Pharmaceutical formulation, serum, urine	[284]
	Phenylalanine	2.18 nM/0.06–25.5 µM	99.24–99.90%	
	Chlorpheniramine	3.63 nM/0.045–242 µM		
	Dextromethorphan	9.10 nM/0.062–242 µM		
MWCNTs	Pioglitazone	Potentiometry	Tablet	[285]
		0.6 µM/1µM–10 mM	99.72–101.12%	
Graphene quantum dots	Isoproterenol	DPV	Ampoule, urine	[286]
		0.6 µM/1–900 µM	98–103.4%	
Gd_2_O_3_ NPs	Venlafaxine	DPV	Tablet, urine, water	[287]
		0.21 µM/5–900 µM	98–103.3%	
Molecularly imprinted polymer NPs/graphene	Sertraline	SWV	Tablet, serum	[288]
		1.99 nM/5–750 nM	97.98–101.33%	

## Data Availability

Not applicable.

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
