# Peer review of "Metal Nanoparticles and Carbon-Based Nanomaterials for Improved Performances of Electrochemical (Bio)Sensors with Biomedical Applications"

_materials, 2021, doi:10.3390/ma14216319_

Round 1
Reviewer 1 Report
This work presents "Nanostructured materials for improved performances of electrochemical (bio)sensors with biomedical applications". The authors reviewed metal nano particles and carbon nanomaterials-based electrochemical biosensors for biomedical applications. The topic is current and important for academic research areas. This review is recommended to be published after including and addressing the below listed comments with major corrections.
- The authors should eliminate the current grammatical and punctuation mark errors and also confirm the correct scientific English.
- The authors should write the complete terms of all abbreviations before the first use in the abstract and main manuscript, figure caption, and tables.
- The authors should clearly explain the innovation and importance of their work on the introduction and highlight the novelty of the review. They should justify the value of the work and compare their work with previously similar published review papers. The introduction section needs to be elaborated.
- The authors should cite important and recent references related to the topic. The below review papers and manuscript should be cited in the revised manuscript:
Sensors 20 (13), 3675 (2020)
RSC Advances 11 (10), 5411-5425 (2021)
RSC Advances 11 (5), 3049-3057 (2021)
Journal of Food Measurement and Characterization 15, 3837–3852 (2021)
- The number of figures (three) for a review paper is not enough. The current figures do not present any specific information to the readers. The quality of the figures must be improved. If the figures are reproduced from other papers, it must be mentioned in the manuscript and the figure captions.
-The caption of the figures should briefly explain the key idea of the figure.
- The authors should include some figures which present synthesis and preparation of materials for each group of materials like gold, silver, carbon nanotube, etc. It is suggested that the authors include some figures which present preparation and device fabrication for each group of materials like gold, silver, carbon nanotube, etc.
- Biocompatibility of the materials and sensors should be clearly discussed.
- Advantage and disadvantage of each material and biosensor should be discussed.
- The style of the tables should be improved.
- The authors should add a section for future outlook and future applications.
- The conclusion must be expanded and highlight the key points
Reviewer 2 Report
This review article contains up-to-date information on the preparation and use of biomedical nano-sized noble metal particles, and I have no critical comments about its content.
My comment concerns the copyright of the three drawings presented. For example Figure 1. Is this picture created from pictures published in different articles [7-20]? Is permission not needed?
Reviewer 3 Report
The authors reviewed the recent development of nanomaterials as electrochemical biosensor for bio-detections and real time diagnosis of diseases. The authors mainly focused on metal nanoparticles based materials and carbon based materials, and discussed the combination of these two types of materials in biosensor design. This research area is developed quickly in recent years. A review like this manuscript will be very meaningful to summary the current progress, discuss the recent challenges and foreseen the perspectives. I would suggest this manuscript be accepted after minor revision.
- In section 2, the authors reviewed the metal-nanoparticles based electrochemical biosensors in three main categories, including Au-based nanoparticles, Ag-based nanoparticles, and Pt-based nanoparticles, because these materials are most popular for electrochemical biosensors. However, there are also a large amount of reports about the applications of other metal nanoparticles in this field. So I would still suggest an extra part be added in this section to introduce other metal-NPs-based electrochemical biosensors.
- Besides the metal nanoparticles based materials and carbon based materials, there are other types of nanostructured electrochemical biosensor materials, such as silica based materials, quantum-dot based materials, quaternary chalcopyrite materials, nanowires, polymers, etc. These materials should also be reviewed under this topic.
Round 2
Reviewer 1 Report
Thanks for the revised version. The comments made during the previous review were properly addressed. I think this manuscript is ready to publish.